# Analysis of Medication Adherence and Its Influencing Factors in Patients with Schizophrenia in the Chinese Institutional Environment

**DOI:** 10.3390/ijerph18094746

**Published:** 2021-04-29

**Authors:** Wei Yu, Jie Tong, Xirong Sun, Fazhan Chen, Jie Zhang, Yu Pei, Tingting Zhang, Jiechun Zhang, Binggen Zhu

**Affiliations:** Shanghai Pudong New Area Mental Health Center, Tongji University School of Medicine, 165 Sanlin Road, Shanghai 200124, China; yuwei15000320971@sina.com (W.Y.); tongjie_room@126.com (J.T.); Sunxr@shspdjw.com (X.S.); chenfz@shspdjw.com (F.C.); zhangj@shspdjw.com (J.Z.); Peiy@shspdjw.com (Y.P.); zhangtt@shspdjw.com (T.Z.); zhangjc@shspdjw.com (J.Z.)

**Keywords:** medication adherence, influencing factors, schizophrenia, Chinese, institutional environment

## Abstract

Background: Factors related to medication adherence in patients with schizophrenia have always been key to the treatment and rehabilitation of these patients. However, the treatment modes in different countries are not the same, and there is no research on the factors influencing medication adherence under different mental health service modes. Objectives: The purpose of this study was to explore medication adherence and its influencing factors in patients with schizophrenia in the Chinese institutional environment. Methods: We conducted a cross-sectional study of hospitalized persons living with schizophrenia from November 2018 to January 2019. A systematic sampling method was used to select 217 hospitalized persons living with schizophrenia. The Medication Adherence Rating Scale (MARS), Positive and Negative Syndrome Scale (PANSS), General Self-Efficacy Scale (GSES), Schizophrenia Quality of Life Scale (SQLS), and Scale of Social Skills for Psychiatric Inpatients (SSPI) were used to explore medication compliance and its influencing factors in the Chinese institutional environment. Results: The descriptive analysis and ANOVA showed that there were no significant differences in medication adherence when assessed by demographic characteristics such as sex, marital status, and education level (*p* > 0.05). A correlation analysis showed that there was no significant correlation between medication adherence and mental symptoms (*p* > 0.05) but that there was a positive correlation with self-efficacy, quality of life, and activities of daily living (*p* < 0.01). The linear regression analysis showed that self-efficacy, psychosocial factors, symptoms/side effects, and activities of daily living had significant effects on medication adherence (F = 30.210, *p* < 0.001). Conclusions: Our findings show that the self-efficacy, quality of life, and social function of patients with schizophrenia are important self-factors influencing medication adherence in the Chinese institutional environment.

## 1. Introduction

Schizophrenia is a chronic, often debilitating psychiatric illness [1]. Typically manifesting in late adolescence or early adulthood, schizophrenia can disturb perception, cognition, emotions, and behavior [2,3]. The number of schizophrenic patients increased from 13.1 million in 1990 to 20.9 million in 2016 [4]. In China, the lifetime prevalence of schizophrenia is 0.7%, with 9.1 million people diagnosed with this illness [5]. According to the World Health Organization (WHO), mental illness will account for one-fourth of the total global burden of disease by 2020. Schizophrenia ranks seventh in the global burden of nonfatal diseases, accounting for 2.8% of years lived with disability (YLD) [6].

At present, schizophrenia can be treated by drugs, psychological therapy, physical therapy, and surgery [7,8]. However, among them, drug treatment is still the main treatment for schizophrenia, and the principles are sufficient treatment, full maintenance, and long-term treatment [9]. Morken [10] et al. found that poor medication compliance among schizophrenic patients is common, with a proportion of 30.9–60.3%. Research shows that young women have better medication adherence than men, that elderly patients may be affected by memory impairment, and that patients with poor socioeconomic status and severe mental symptoms also have poor medication adherence [11]. In China, 20% of community patients with schizophrenia were considered non-adherent [12]. However, there is no research on the effect of treatment modes in different countries on the medication adherence of patients with schizophrenia.

To date, the trend in mental health services in Western countries tends to be deinstitutionalization, forming a psychiatric hospital–community integration model [13]. Some countries, such as Italy, have even ended the public psychiatric hospital system and have formed independent community mental health service systems [14]. In different regions of China, institutionalization is still the core of the mental health service system. The number of beds for patients with mental health diseases accounts for 74.9% of the total number of beds allocated, and 43.6% of patients have been hospitalized for more than five years [15]. A multi-center study in 2012 showed that patients with schizophrenia accounted for the largest proportion of long-term inpatients in China’s mental health institutions (78.6%), followed by patients with organic mental disorder (5.6%) and patients with intellectual disabilities (4.6%) [16]. In addition, the Chinese mental health institutional environment is different from that of the West, and more attention is given to antipsychotic treatment [17].

Few studies have focused on the factors influencing medication adherence in schizophrenia under different mental health service environments. The purpose of this study is to explore the differences in medication adherence under different mental health service modes by studying the factors influencing medication adherence of patients with schizophrenia in the Chinese institutional environment.

## 2. Methods

### 2.1. Procedure

A cross-sectional study was performed. The sample size of the study was calculated based on the formula $N_{sample}={\left(\frac{{Ζ}_{1-\alpha /2}\times \sigma }{\delta }\right)}^{2}$ [18]. The significance level was 0.05, and a two-sided test was required. The estimated sample size was 178. The study included at least 223 participants, accounting for an approximately 20% loss to follow-up (those not discharged automatically according to the discharge standards or who could not complete the study for other specific reasons). All participants were from the Shanghai Pudong New Area Mental Health Center and the Tongji University School of Medicine (Shanghai, China). The study lasted from November 2018 to January 2019, and sample collection and scale evaluation were performed by professional clinical researchers. Subsequently, an experimenter instructed the participants to complete the self-reported scales in a quiet room. The experimenter checked the completed scales for blank entries for each item.

### 2.2. Participants

The sample of 807 inpatients was originally recruited from the Shanghai Pudong New Area Mental Health Center and the Tongji University School of Medicine. The following inclusion criteria were employed: (1) a diagnosis of schizophrenia according to the DSM-5 [19] criteria, (2) age ≥ 18 years and ≤70 years, (3) continuous hospital stay ≥one month, (4) education above junior high school, (5) no visual or auditory impairment, (6) capacity to complete the self-test scale independently, (7) participant/legal guardian/next of kin provision of written informed consent, and (8) a Positive and Negative Syndrome Scale (PANSS) score ≤ 60. A total of 652 patients met the DSM-5 diagnostic criteria for schizophrenia, accounting for 80.79% of all hospitalized patients. Forty-three patients had affective disorders, accounting for 5.33% of all hospitalized patients. Thirty patients had an intellectual disability, accounting for 3.72% of all hospitalized patients. Finally, 316 patients with schizophrenia (39.16% of all hospitalized patients) met the inclusion criteria (see Figure 1). A systematic sampling method was used to select 223 patients who met the inclusion criteria. Six patients could not complete all of the tests effectively because of temporary discharge, aggravation, or an inability to cooperate, and the loss to follow-up rate was 2.69%.

### 2.3. Ethics

The study was approved by the Ethics Committee of the Shanghai Pudong New Area Mental Health Center and the Tongji University Mental Health Center (No: 201611). All procedures were performed in accordance with the ethical standards of the responsible committee on human experimentation (institutional and national) and the 1975 Declaration of Helsinki, as revised in 2008. All of the enrolled participants had sufficient reading proficiency to understand and to complete the questionnaires, and to provide written informed consent.

## 3. Measures

The purpose of this study was to explore medication adherence and its influencing factors in patients with schizophrenia under Chinese institutionalization. The study applied both qualitative and quantitative assessments of patient responses using the MARS, PANSS, GSES, SQLS, and SSPI effectively to draw out more nuanced results of the related factors to medication compliance.

### 3.1. Medication Adherence Rating Scale (MARS)

The Medication Adherence Rating Scale (MARS) is suitable for quantitative evaluation of drug compliance in patients with schizophrenia. The MARS was compiled in 2000 by a psychometrist, Dr. Thompson, with Mental Health Services for Kids and Youth, Australia [20]. The Chinese version of the MARS was introduced by Professor Yucheng Kao in 2010. The scale measures the following 10 items: (1) Do you ever forget to take your medicine? (2) Are you careless at times about taking your medicine? (3) When you feel better, do you sometimes stop taking your medicine? (4) Sometimes if you feel worse when you take your medicine, do you stop taking it? (5) I take my medication when I am sick. (6) It is unnatural for my mind and body to be controlled by medication. (7) My thoughts are clearer on medication. (8) By staying on medication, I can prevent getting sick. (9) I feel weird, like a ‘zombie’, on medication. (10) Medication makes me feel tired and sluggish. Except for items 7 and 8, the answer ‘yes’ was scored as 1 point and the answer ‘no’ was scored as 0 points. The lower the total score is, the better the medication adherence is. The MARS is an effective and reliable tool for the quantitative evaluation of medication adherence in schizophrenia patients. The scale was completed by patients themselves. Cronbach’s α coefficient for the scale was 0.800 [21].

### 3.2. Positive and Negative Syndrome Scale (PANSS)

The Positive and Negative Syndrome Scale (PANSS) was used to evaluate the presence and severity of mental symptoms. The PANSS was compiled by Professor Stanley R. Kay in 1987, a psychometrist and psychological researcher at the Albert Einstein College of Medicine, USA [22]. The Chinese version of the PANSS was introduced in China by Professor Yanling He in 1997. The scale was evaluated with 30 basic items and consisted of 3 subscales. There were 6 items on the positive scale, 6 items on the negative scale, and 16 items on the general psychopathology scale. Each item on the PANSS is accompanied by a complete definition as well as detailed anchoring criteria for all seven rating points, which represent increasing levels of psychopathology: 1 = absent, 2 = minimal, 3 = mild, 4 = moderate, 5 = moderate-severe, 6 = severe, and 7 = extreme. A higher total score indicates a higher degree of psychopathology. The PANSS is an effective and reliable tool for the quantitative evaluation of symptoms in patients with schizophrenia. The scale was evaluated by trained professionals. Cronbach’s α coefficient for the scale was 0.870 [23].

### 3.3. General Self-Efficacy Scale (GSES)

The General Self-Efficacy Scale (GSES) was used to evaluate the self-efficacy of the patients. The GSES was compiled by Professor Ralk Schwarzer in 1981, a clinical psychologist at the Free University of Berlin, Germany [24]. The Chinese version of the GSES was introduced in China by Professor Jianxin Zhang in 1995 [25]. The scale measures the following 10 items: (1) If I try my best, I can always solve problems. (2) I can still get what I want even though others are against me. (3) It is easy for me to stick to my ideals and to achieve my goals. (4) I have confidence that I can deal with unexpected events effectively. (5) With my intelligence, I can cope with unexpected situations. (6) When I make the necessary efforts, I can solve most problems. (7) I can face difficulties calmly because I trust my ability to deal with problems. (8) When facing a difficult problem, I can usually find several solutions. (9) When I experience challenges, I usually think of ways to deal with them. (10) No matter what happens to me, I can handle it easily. Using Likert’s 4-grade scoring method, “totally incorrect” is scored with 1 point, “somewhat correct” is scored with 2 points, “mostly correct” is scored with 3 points, and “completely correct” is scored with 4 points. The total score ranges from 10–40 points. A higher total score indicates a higher degree of self-efficacy. The GSES is an effective and reliable tool for the quantitative evaluation of the self-efficacy of patients with schizophrenia. The scale was completed by patients themselves. Cronbach’s α coefficient for the scale was 0.831 [26].

### 3.4. Schizophrenia Quality of Life Scale (SQLS)

The Schizophrenia Quality of Life Scale (SQLS) was used to evaluate the quality of life of patients with schizophrenia. The SQLS was compiled by Professor G. Wilkinson in 1999, a clinical psychiatrist at the University of Liverpool, UK [27]. The Chinese version of the SQLS was introduced in China by Professor Jie Li in 2001. The scale measures the following 30 items: (1) I lack the energy to do things. (2) I am bothered by my shaking/trembling. (3) I feel unsteady walking. (4) I feel angry. (5) I am troubled by a dry mouth. (6) I can’t be bothered to do things. (7) I worry about my future. (8) I feel lonely. (9) I feel hopeless. (10) My muscles get stiff. (11) I feel very jumpy and edgy. (12) I am able to carry out my day-to-day activities. (13) I take part in enjoyable activities; (14) I take things people say the wrong way. (15) I like to plan ahead. (16) I find it hard to concentrate. (17) I tend to stay at home. (18) I feel it is difficult to mix with people. (19) I feel down and depressed. (20) I feel like I can handle something. (21) My vision is blurred. (22) I feel very mixed up and unsure of myself. (23) My sleep is disturbed. (24) My feelings go up and down. (25) I get muscle twitches. (26) I am concerned that I won’t get better. (27) I worry about things. (28) I feel that people tend to avoid me. (29) I get upset thinking about the past. (30) I get dizzy spells. The SQLS has good reliability and validity and is used to assess the quality of life of patients with schizophrenia. Respondents could select a response to each question from ‘Never’ (0), ‘Rarely’ (1), ‘Sometimes’ (2), ‘Often’ (3), or ‘Always’ (4). The scale consists of three subscales: psychosocial, motivation and energy, and symptoms and side effects. Each scale is scored from 0 to 100, with 0 indicating the best quality of life and 100 indicating the worst quality of life. The scale was completed by patients themselves. Cronbach’s α coefficient for the scale was 0.930 [27].

### 3.5. Scale of Social Skills for Psychiatric Inpatients (SSPI)

The Scale of Social Skills for Psychiatric Inpatients (SSPI) was used to evaluate the social functions of the patients. The SSPI was compiled by Professor Chaodang Zhou in 2003. The SSPI includes 12 items in total, including factor I: daily living ability, factor II: mobility and communication, and factor III: social activity skills. Using Likert’s 5-grade scoring method, ‘a lack of this function’ is scored with 0 points; ‘the need to expend considerable energy to complete a project’ is scored with 1 point; ‘the existence of this function, but only under supervision’ is scored with 2 points; ‘the ability to complete a project independently, but with little enthusiasm or initiative’ is scored with 3 points; and ‘the ability to perform well consistently’ is scored with 4 points. The total score ranges from 0–408 points. Higher total scores indicate stronger social function. The SSPI is an effective and reliable tool for quantitative evaluation of the social function of patients with schizophrenia. The scale was evaluated by trained professionals. Cronbach’s α coefficient for the scale was 0.871 [28].

The PANSS and SSPI were assessed by 5 psychiatrists who received unified training. The Kappa value was set to 0.83 for consistency (*p* > 0.05). All evaluators had a master’s degree in psychiatry and were licensed Chinese psychometrists. The MARS, GSES, and SQLS were completed by the patients themselves.

### 3.6. Statistical Analysis

All statistical analyses of the data were performed using the SPSS 20.0 software package (SPSS, Chicago, IL, USA). The descriptive analysis and analysis of variance examined the demographic characteristics and medication adherence. The Kolmogorov–Smirnov one-sample test was used to assess whether the MARS scores were normally distributed. A correlation analysis was used to explore the correlation between medication adherence, positive and negative syndrome, general self-efficacy, and quality of life. Linear regression analysis was used to explore the relationship between significant factors and medication adherence. The associations between variables were considered significant at a level of *p* < 0.05.

## 4. Results

### 4.1. Sample Profile

There were no significant differences between the sex, marital status, or education level and the medication adherence of the participants (*p* > 0.05). There were 110 males (50.96%) and 107 females (49.04%). There were 147 married (68.27%), 48 unmarried (22.12%), and 22 divorced or widowed (9.61%) participants. A total of 121 (55.77%) had an education level of junior high school or below, 67 (30.77%) had a senior high school level of education, and 29 (13.46%) had a junior college or above level of education (see Table 1).

### 4.2. Correlation Analysis of Medication Adherence

As Table 2 shows, there was no significant correlation between MARS scores and PANSS factor scores (positive symptoms, negative symptoms, general psychopathology, and aggressive risk) (*p* > 0.05). There was a positive correlation between MARS scores and GSES scores (*p* < 0.01). There was a negative correlation between MARS scores and SQLS factor scores (psychosocial, energy/motivation, and symptoms/side effects) (*p* < 0.01). There was a positive correlation between MARS scores and SSPI ‘ability of daily living’ factor scores (*p* < 0.01), but there was no significant correlation with other factors of the SSPI (*p* > 0.05).

### 4.3. Linear Regression Analysis of Medication Adherence

As Table 3 shows, the linear regression analysis was carried out by taking the MARS score as the dependent variable, and the significant factors in the univariate analysis and those with correlations in the correlation analysis as the independent variables. Self-efficacy, psychosocial factors, symptoms/side effects, and activities of daily living had a significant effect on medication adherence (F = 30.210, *p* < 0.001). The results showed that medication adherence was positively correlated with self-efficacy and activities of daily living and negatively correlated with psychosocial factors and symptoms/side effects.

## 5. Discussion

Most studies have found that demographic characteristics have an impact on medication adherence in patients with schizophrenia. A study of 393 schizophrenic patients by Eticha [29] found that 26.5% were nonadherent to their antipsychotic medication. Among them, the medication adherence of young patients was lower than that of older patients and education level was positively correlated with medication adherence. Moritz [30] et al. found that the medication adherence of female patients was higher than that of male patients and that of married patients was higher than that of unmarried patients. Sweileh [31] et al. found that patients with a short course of disease had relatively poor medication adherence. However, our study showed that there was no correlation between the demographic characteristics and medication adherence of patients with schizophrenia, which is related to drug treatment being the core of schizophrenia therapy under Chinese institutionalization.

This study found that there was no significant correlation between medication adherence and mental symptoms, including positive symptoms, negative symptoms, general psychopathology, and aggressive risk. This result is also different from that of the noninstitutional environment. Yang [32] et al. found that mental symptoms had a significant impact on medication adherence in patients with schizophrenia. Ansari [33] et al. showed through a study of 100 schizophrenic patients that patients with high PANSS scores had poor compliance, that patients with delusions of victimization refused to take drugs, and that patients with negative symptoms were affected by compliance behavior because of a lack of motivation. This finding may be related to the long-term hospitalization pattern of patients in the context of the Chinese institutional environment, which is mainly caused by negative symptoms and selection bias in symptoms.

Mental illness has a long course and easily relapses, and patients often worry that others do not understand their medication. Therefore, patients easily lack sufficient confidence in treatment [34]. At present, there is no research on the self-efficacy and medication adherence of patients with schizophrenia; there is only research on the correlation between self-efficacy and mental symptoms [35]. Through correlation analysis and regression analysis, our study revealed that there was a positive correlation between self-efficacy and the medication adherence of patients with schizophrenia, and self-efficacy is an important influencing factor of medication adherence. Therefore, strengthening patients’ awareness of the disease and increasing their self-efficacy and treatment confidence can effectively improve their medication adherence.

Side effects are one of the most common causes of drug noncompliance. Tabea [36] et al. found that 37.7% of patients were worried that long-term medication would affect their work, marriage, and health, leading to, for example, obesity, stupidity, memory loss, dizziness, fatigue, and mental depression; therefore, they did not accept long-term maintenance medication. Our study also confirmed the correlation between side effects and medication adherence and found that psychosocial factors in quality of life also have an important impact on the drug compliance of patients. It is of great significance to choose drugs with relatively mild side effects and to improve patient quality of life, especially psychosocial factors, to increase medication adherence [37].

Social function has always been an important standard to measure the rehabilitation of patients with schizophrenia [38]. Because of the influence of antipsychotic drugs, long-term hospitalization and the evolution of the disease itself, the negative symptoms and social function of patients decreased, which accelerated mental decline [39,40]. Our study found that social function, especially activities of daily living, plays an important role in drug compliance in patients with schizophrenia. It has been suggested that psychological and mental rehabilitation therapy not only can slow down the decline of social function but also can improve medication adherence and the mental symptoms of patients [41].

### Limitations and Recommendations for Future Studies

Our study has several limitations that provide directions for future research. First, the present research is limited to patients with schizophrenia in the Chinese institutional environment. In the future, we may be able to compare the differences between the factors influencing medication adherence in other countries with those in non-institutionalized environments. Second, this study is limited by linguistic issues given that all data were collected in Chinese and were translated into English by the researcher. This process might have resulted in the participants’ statements being conveyed differently from what they had originally meant to say. Third, the study is only a cross-sectional study. In the future, large sample, multi-center, randomized controlled trials can be conducted to explore the effects of different diseases, such as affective disorder or intellectual disability, administration methods, drug types, and intakes on medication adherence.

## 6. Conclusions

This study found that the self-efficacy, quality of life, and social function of patients with schizophrenia are important self-factors influencing medication adherence in the Chinese institutional environment. However, there was no correlation between medication compliance and demographic characteristics or mental symptoms. Medication adherence was positively correlated with self-efficacy and activities of daily living and negatively correlated with psychosocial factors and symptoms/side effects. The difference between the results of this study and those from research on factors influencing medication adherence in other countries may be related to the hospitalization mode of patients with schizophrenia in the context of the Chinese institutional environment.

## Figures and Tables

**Figure 1 ijerph-18-04746-f001:**
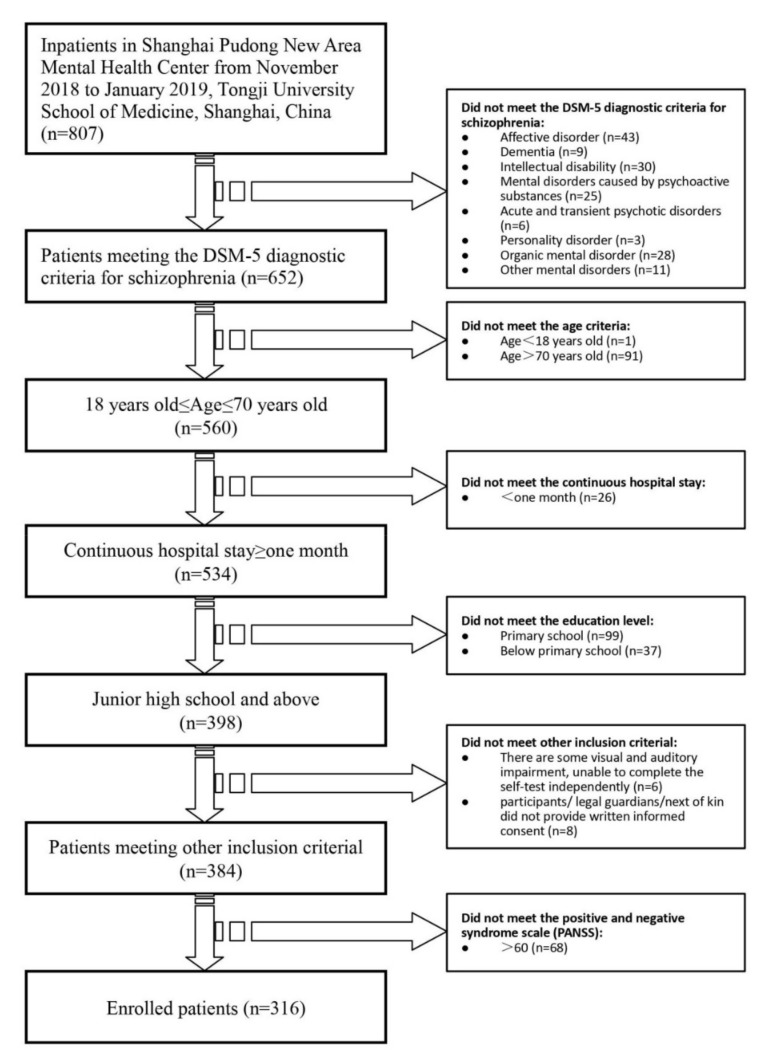
A cross-sectional study path diagram.

**Table 1 ijerph-18-04746-t001:** Comparison of demographic characteristics and medication adherence of the participants.

Variable	Number (%)	MARS
Gender (sample)		*p* = 0.27
Male	110 (50.96%)	5.26 ± 2.89
Female	107 (49.04%)	4.87 ± 2.58
Marital status (sample)		*p* = 0.07
Married Single	147(68.27%)	5.31 ± 2.36
Single	48 (22.12%)	3.96 ± 2.55
Divorce or widowhood	22(9.61%)	5.20 ± 2.74
Education level (sample)		*p* = 0.79
Junior high school or below	121 (55.77%)	4.88 ± 2.46
Senior high school	67 (30.77%)	5.25 ± 2.50
Junior college or above	29 (13.46%)	4.93 ± 2.64

Data are presented as means.

**Table 2 ijerph-18-04746-t002:** Correlation analysis of medication adherence of patients with schizophrenia.

Variable	*r*	*p*
PANSS scale		
Positive symptoms	−0.11	0.26
Negative symptoms	−0.04	0.71
General psychopathology	−0.05	0.60
Aggressive risk	−0.10	0.31
GSES scale		
Self-efficacy	0.21	<0.01 ***
SQLS scale		
Psychosocial	−0.71	<0.01 ***
Energy/motivation	−0.48	<0.01 ***
Symptoms/side effects	−0.70	<0.01 ***
SSPI scale		
Activities of daily living	0.36	<0.01 ***
Mobility and communication	0.17	0.09
Social activity skills	0.13	0.21

* *p* < 0.01.

**Table 3 ijerph-18-04746-t003:** Linear regression analysis of medication adherence of patients with schizophrenia.

Variable	*β*	SEM	*t*	*p*	*R* ^2^	∆*R*^2^	*F*
Constant	4.535	1.282	3.536	0.011	0.607	0.586 *	30.210 **
Self-efficacy	0.068	0.028	2.390	0.019			
Psychosocial	−0.083	0.024	−3.460	0.001			
Energy/motivation	0.034	0.059	0.567	0.572			
Symptoms/side effects	−0.160	0.041	−3.912	0.000			
Activities of daily living	0.209	0.097	2.162	0.033			

* *p* < 0.01. ** *p* < 0.001.

## Data Availability

The data presented in this study are available on request from the corresponding author.

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
