# Peer review of "Analysis of Medication Adherence and Its Influencing Factors in Patients with Schizophrenia in the Chinese Institutional Environment"

_ijerph, 2021, doi:10.3390/ijerph18094746_

Round 1

Reviewer 1 Report

This paper is aimed to a very important items, the drug compliance in psychiatric patients, mainly exploring the differences of medication adherence under different mental health service models by studying the influencing factors of medication adherence of patients with schizophrenia under Chinese institutional environment.

The paper explores a selected inpatients population in a metropolitan area within China, so result con be referred only to this geographical methodological and temporal setting.

The main concern of the work is that the assessment of adherence to therapy is evident from the statements of the subjects and the completion of the test. So far, there are no objective measures of the assumption of pills, dosage of the substance in the blood or urine or other objective methods of verifying drug intake, it is very difficult to attribute a predictive and objective value to the results.

The consistency of the results that correlate compliance with the side effects confirms the subjectivity of the response. If the patients were all hospitalized, how could they not take the drugs. The method of administration of the medicines and the assessment of intake must be described.

The work is interesting, the purpose very useful for the evaluation of the effectiveness of interventions in this field, but without an objective measurement of drug assumption it is not acceptable for publication in an international journal.

Bibliography

Some examples:

Medication Adherence Decisions in Patients With Schizophrenia. Kikkert MJ, Dekker J.

Prim Care Companion CNS Disord. 2017 Dec 7;19(6):17n02182. doi: 10.4088/PCC.17n02182.

PMID: 29216418 Free article. Review.

Antipsychotic Drugs in Schizophrenia: Relative Effects in Patients With and Without Treatment Resistance.

Andrade C.  J Clin Psychiatry. 2016 Dec;77(12):e1656-e1660. doi: 10.4088/JCP.16f11328.

PMID: 28086018 Free article. Review.

Medication adherence in patients with schizophrenia. Phan SV.

Int J Psychiatry Med. 2016;51(2):211-9. doi: 10.1177/0091217416636601.

PMID: 27079779 Review.

Optimizing Treatment Choices to Improve Adherence and Outcomes in Schizophrenia. Kane JM, Correll CU.

J Clin Psychiatry. 2019 Sep 17;80(5):IN18031AH1C. doi: 10.4088/JCP.IN18031AH1C.

PMID: 31536686 Free article.

Medication adherence in patients with schizophrenia: a qualitative study of the patient process in motivational interviewing. Dobber J, Latour C, de Haan L, Scholte Op Reimer W, Peters R, Barkhof E, van Meijel B.

BMC Psychiatry. 2018 May 18;18(1):135. doi: 10.1186/s12888-018-1724-9.

PMID: 29776393 Free PMC article.

Compliance in schizophrenia spectrum disorders: the role of clinical pharmacist. Yalçin N, Ak S, Gürel ŞC, Çeliker A.

Int Clin Psychopharmacol. 2019 Nov;34(6):298-304. doi: 10.1097/YIC.0000000000000280.

PMID: 31343497

Author Response

Dear Professor,

Thank you very much for your review and suggestions.

I've revised the manuscript one by one. The English language and style has been revised by native English speaking experts from MDPI. I really appreciate your help. The problems are as follows:

Q1: This paper is aimed to a very important items, the drug compliance in psychiatric patients, mainly exploring the differences of medication adherence under different mental health service models by studying the influencing factors of medication adherence of patients with schizophrenia under Chinese institutional environment.

The paper explores a selected inpatients population in a metropolitan area within China, so result con be referred only to this geographical methodological and temporal setting.

  • In the introduction, it is added that the current management mode of schizophrenics in different regions of China is mainly under the institutional environment, which is different from that in western countries. In the limitations, it is added that we could try to explore the related factors of medication adherence through large sample, multi center and randomized controlled trials in the future.

Q2: The main concern of the work is that the assessment of adherence to therapy is evident from the statements of the subjects and the completion of the test. So far, there are no objective measures of the assumption of pills, dosage of the substance in the blood or urine or other objective methods of verifying drug intake, it is very difficult to attribute a predictive and objective value to the results.

  • Thank you very much for the verifying drug intake and other factors proposed by professor, which is also the direction of future research. The drug factor has been added to the Limitations and recommendations for future studies.

Q3: Because this study focuses on the influencing factors of medication compliance of patients under a certain mental health management system, the future research direction is added in the restriction part, which is to study the medication compliance of patients with schizophrenia by different administration methods, drug types and intakes.

  • Because the study focuses on the subjective factors of medication adherence of patients with schizophrenia under Chinese institutional environment, which is different from that of western countries. In the Limitations, it is added that the research is to study the compliance of different administration methods, drug types and intake of patients with schizophrenia in the future.

Reviewer 2 Report

Abstract

  1. Please specify the study period in the methods section.
  2. What statistical analyses did you perform in the study? What were your outcome variables? Please specify them in the methods section.
  3. Line 33: It is great significance to reduce the recurrence of disease, improve the quality of life of patients, reduce medical costs and promote the prognosis of patients.

→ The study findings did not support this argument. Please revise it.

Introduction

  1. Line 42: The reference 3 is out of date and does not support your argument.
  2. Line 50-52: Medication adherence is important for people with any diseases, not schizophrenia alone. I don’t know why it is important to address schizophrenia. What is the rate of medication adherence among people with schizophrenia in China? Your argument is not reasonable.
  3. You need to do a thorough literature review to address why this research question is important. What were the findings of prior research about the medication adherence sorresponding to its associated factors? Why the stuies from other countries can’t be applied to the contexts in China? What is the existing knowledge and gap of the moderation adherence among people with schizophrenia? Your current introduction is too weak to show the value of this study, and you need to strengthen it.

Methods

  1. How did you identify and approach potential participants and provide informed consent?
  2. Line 116: The reference 19 did not match your statement.
  3. Line 123-124: Except for items 7 and 8, the answer to "yes" was scored 1 point, and the 123 answer to "no" was scored 1 point.

→ Your information is conflicting, please check the accuracy and correct it.

  1. As you mentioned the MARS has a good validity, what the validity of the MARS was?
  1. Line 133: The reference 22 did not match your statement.
  2. As you mentioned the PANSS has a good validity, what the validity of the PANSS was?
  3. As you mentioned the GSES has a good validity, what the validity of the GSES was?
  4. Line 166: The reference 28 did not match your statement.
  5. As you mentioned the SQLS has a good validity, what the validity of the SQLS was?
  6. As you mentioned the SSPI has a good validity, what the validity of the SSPI was?
  7. What theoretical framework did you use to inform the study design and variable selection?
  8. How did you calculate and decide the sample size was large to have sufficient power?
  9. What were the power and effect size of this study?
  10. How did you use ANOVA in this study? Which variables did you use ANOVA for analyses?
  11. What were your outcome and independent varianbles? What were the variables that were placed as covariates in the regression model? What was your rationale to select covariates?
  12. You don’t need to perform t-test or ANOVA. Using regression after controlling for covariates can answer your question.

Results

  1. Age and course of disease are continuous variables in nature. Please use mean and standard deviation to present them but not treat them as dichotomous variables.
  2. Please provide a table to describe bivcariate correlation matrix among variable.
  1. You need to test the model by controlling for all sociodemographics. Please rerun the mode testing and report results, accordingly.
  2. Please provide the standardized regression coefficient in the results section.

Discussion

  1. A lot of information in the discussion section is self-evident but not supported by the study findings. You need to look into your study findings to provide meaningful information closely linking to the research questions.
  1. There were ample limitations of study design that should be mentioned, such as sampling strategy, participant selection, misinformation, etc. Please provide more information in this section.
  2. What is the uniqueness generated from your study? Your results were repetitive of the findings of studies from other countries. In addition, the study was conducted in the hospital alone, but can not anser your research question that aimed to understand the situation in different settings.What are your suggestions regarding implementations for daily practice and future study? You have to compare your findings to the existing literature and discuss more information relevant to your research question and provide concrete suggestions to enrich existing knowledge of the care for people with schizophenria.

Conclusion

  1. Line 315-318 and line 320-322: The study findings did not support these arguments. Please revise them in a proper manner.

Author Response

Dear Professor,

Thank you very much for your review and suggestions.

  • I've revised the manuscript one by one. The English language and style has been revised by native English speaking experts from MDPI. I really appreciate your help. The problems are as follows:

Q1: Abstract

Please specify the study period in the methods section.

  • The study period has been added in the methods of abstract

What statistical analyses did you perform in the study? What were your outcome variables? Please specify them in the methods section.

  • Specific statistical methods have been added in the abstract.

Line 33: It is great significance to reduce the recurrence of disease, improve the quality of life of patients, reduce medical costs and promote the prognosis of patients. The study findings did not support this argument. Please revise it.

  • Specific statistical methods have been added in the abstract.

The sentence of “It is great significance to reduce the recurrence of disease, improve the quality of life of patients, reduce medical costs and promote the prognosis of patients” in the abstract has been deleted.

Q2: Introduction

Line 42: The reference 3 is out of date and does not support your argument.

  • Line 42 and the reference 3 has been deleted.

Line 50-52: Medication adherence is important for people with any diseases, not schizophrenia alone. I don’t know why it is important to address schizophrenia. What is the rate of medication adherence among people with schizophrenia in China? Your argument is not reasonable.    

  • Line 50-52 has been deleted. The importance of medication adherence to schizophrenia, the rate of medication adherence in China and etc has been added.

Q3: Methods

How did you identify and approach potential participants and provide informed consent?

  • The method of sample size selection has added in the methods. The informed consent of the participants has been added in the ethics.

Line 116: The reference 19 did not match your statement.

  • The reference has been deleted. Because only indirect references can be found in English literature. At present, there are only Chinese versions of direct reference.

Line 123-124: Except for items 7 and 8, the answer to "yes" was scored 1 point, and the 123 answer to "no" was scored 1 point.

Your information is conflicting, please check the accuracy and correct it.

  • Line 123-124 has been corrected.

As you mentioned the MARS has a good validity, what the validity of the MARS was?

  • The description has been modified.

Line 133: The reference 22 did not match your statement.

  • The reference has been deleted. Because only indirect references can be found in English literature. At present, there are only Chinese versions of direct reference.

As you mentioned the PANSS has a good validity, what the validity of the PANSS was?

  • The description has been modified.

As you mentioned the GSES has a good validity, what the validity of the GSES was?

  • The description has been modified.

Line 166: The reference 28 did not match your statement.

  • The reference has been deleted. Because only indirect references can be found in English literature. At present, there are only Chinese versions of direct reference.

As you mentioned the SQLS has a good validity, what the validity of the SQLS was?

  • The description has been modified.

As you mentioned the SSPI has a good validity, what the validity of the SSPI was?

  • The description has been modified.

What theoretical framework did you use to inform the study design and variable selection?

  • The theoretical framework has been added.

How did you calculate and decide the sample size was large to have sufficient power?

  • The methods of sample size estimation and participants selection has been added in the Methods

How did you use ANOVA in this study? Which variables did you use ANOVA for analyses?

  • The motheds and variables of ANOVA have been added.

What were the variables that were placed as covariates in the regression model? What was your rationale to select covariates?

  • Covariates in regression model are items that have no statistical significance in correlation analysis, such as PANSS, mobility and communication and social activity skills.

You don’t need to perform t-test or ANOVA. Using regression after controlling for covariates can answer your question.

  • The t-test has been deleted.

Q4: Results

Age and course of disease are continuous variables in nature. Please use mean and standard deviation to present them but not treat them as dichotomous variables.

You need to test the model by controlling for all sociodemographics. Please rerun the mode testing and report results, accordingly.

Please provide the standardized regression coefficient in the results section.

  • Considering the unity of the data, the nature variable of age and course of disease has been deleted. The relevant data has been modified. The standardized regression coefficient has been added.

Q5: Discussion

A lot of information in the discussion section is self-evident but not supported by the study findings. You need to look into your study findings to provide meaningful information closely linking to the research questions.

  • The discussion has been improved by making major revisions.

There were ample limitations of study design that should be mentioned, such as sampling strategy, participant selection, misinformation, etc. Please provide more information in this section.

  • The sampling strategy, participant selection, misinformation has been added in the methods.

What is the uniqueness generated from your study? Your results were repetitive of the findings of studies from other countries. In addition, the study was conducted in the hospital alone, but can not anser your research question that aimed to understand the situation in different settings.What are your suggestions regarding implementations for daily practice and future study? You have to compare your findings to the existing literature and discuss more information relevant to your research question and provide concrete suggestions to enrich existing knowledge of the care for people with schizophenria.

  • The highlights of this study have been added in the discussion, focusing on the influencing factors of institutional environment on patients' medication Adherence, and listing the differences of studies in different countries. In addition, the future research direction has been added in the limitation.

Q6: Conclusion

Line 315-318 and line 320-322: The study findings did not support these arguments. Please revise them in a proper manner.

  • The sentence of Line 315-318 and line 320-322 in the conclusion has been deleted and revised.

Reviewer 3 Report

This study aims to explore the differences in medication adherence under different mental health service modes by studying the influencing factors of medication adherence of patients with schizophrenia under Chinese institutional environment. This study found that there was no significant correlation between medication adherence and mental symptoms, including positive symptoms, negative symptoms, general psychopathology, and aggressive risk.

The introduction section introduces sufficiently the thematic, even if it could be shortened. Moreover, I suggest improving the aims of the study. Particularly, in an experimental study, the aims should be clearly written: indeed, the goals should be SMART (Specific, Measurable, Attainable, Relevant, and Timely).

The Materials and Methods section could be linked with section 3, considering that the section Measures is a description of the methods used for the study. In this section, I suggest clarifying only section 3.6, improving the description of the statistical tests.

The results section reported the main findings of the study.

The discussion section should be improved, compare their data with the international literature, while they predominantly summarized their findings.

I appreciate the choice to insert the limitations of the study.

In summary, I suggest improving the manuscript by making minor revisions.

Author Response

Dear Professor,

Thank you very much for your review and suggestions.

I've revised the manuscript one by one. The English language and style has been revised by native English speaking experts from MDPI. I really appreciate your help. The problems are as follows:

Q1: The introduction section introduces sufficiently the thematic, even if it could be shortened. Moreover, I suggest improving the aims of the study. Particularly, in an experimental study, the aims should be clearly written: indeed, the goals should be SMART (Specific, Measurable, Attainable, Relevant, and Timely).

  • The introduction has been shortened and the aims of the study have been improved.

Q2: The Materials and Methods section could be linked with section 3, considering that the section Measures is a description of the methods used for the study. In this section, I suggest clarifying only section 3.6, improving the description of the statistical tests.

  • The materials, Methods and description of the statistical tests have been modified.

Q3: The discussion section should be improved, compare their data with the international literature, while they predominantly summarized their findings.

  • The discussion has been modified, and comparison with the international literature has been added.

Q4: In summary, I suggest improving the manuscript by making minor revisions.

The manuscript has been improved by making minor revisions.

Reviewer 4 Report

The work presented is of enormous interest and relevance. It is worth paying attention to the factors that influence medication adherence in patients with schizophrenia. This is one of the most complicated aspects in the therapeutic management of people with schizophrenia. 

Below we present a series of aspects to be taken into account in order to improve the work presented:
- It would be interesting to revise the abstract and adapt it to a classic format: introduction, objectives, methodology, etc.

- In the introduction the authors consider that the treatment of schizophrenia is still based on antipsychotics but do not take into account other types of non-pharmacological treatments that are currently essential.

- In the paragraph in which the objective is included, the authors have incorporated a series of "collateral benefits" to the work carried out. It would be interesting if these aspects were in the discussion or conclusion.

- In the methodology, the authors incorporate ethical aspects in the "procedure". This section should include the type of study carried out, the literature that supports it and a separate section on ethical aspects. 

- How were the participants in the study selected?

Author Response

Dear Professor,

Thank you very much for your review and suggestions.

I've revised the manuscript one by one. The English language and style has been revised by native English speaking experts from MDPI. I really appreciate your help. The problems are as follows:

Q1: It would be interesting to revise the abstract and adapt it to a classic format: introduction, objectives, methodology, etc.

-  The format of the abstract has been modified to add the objectives, methods, Results.

Q2: In the introduction the authors consider that the treatment of schizophrenia is still based on antipsychotics but do not take into account other types of non-pharmacological treatments that are currently essential.

-  In the secnd paragraph of the introduction, non-pharmacological treatments of schizophrenia has been added.

Q3: In the paragraph in which the objective is included, the authors have incorporated a series of "collateral benefits" to the work carried out. It would be interesting if these aspects were in the discussion or conclusion.

-  The " collateral benefits " in the paragraph in which the objective is included has been deleted and added to the conclusions.

Q4: In the methodology, the authors incorporate ethical aspects in the "procedure". This section should include the type of study carried out, the literature that supports it and a separate section on ethical aspects.

-  The ethics in the "procedure" has been deleted and described in a separate section.

Q5: How were the participants in the study selected?

-  The methods of sample size estimation and participants selection has been added in the Methods.

Round 2

Reviewer 1 Report

In this new version the article could be published in International Journal od Enviromental Research and Public Health.

Reviewer 2 Report

The manuscript has been improved though the findings are not novel.

Reviewer 4 Report

The modifications made by the authors satisfy the requirements made by this reviewer.